# Women's experiences of over-the-counter and prescription medication during pregnancy in the UK: findings from survey free-text responses and narrative interviews

Julia Sanders,[1] Rebecca Blaylock ![ORCID] ,[2] Caitlin Dean,[3] Irene Petersen,[4] Heather Trickey,[5] Clare Murphy[2]

Dr Trickey died during the writing of this manuscript.

JS and RB are joint first authors.

**Correspondence to**
Rebecca Blaylock;
rebecca.blaylock@bpas.org

## ABSTRACT

**Objectives** To explore women's experiences of over-the-counter and prescription medication advice and use during pregnancy.

**Design** A study design consisting of an online survey and nested in-depth interviews with a subsample of participants. We analysed data from survey free-text responses and in-depth interviews using thematic analysis. Quantitative survey data is published elsewhere.

**Setting** The UK.

**Participants** Women were eligible if living in the UK, aged 16–45 years, were pregnant or had been pregnant in the last 5 years regardless of pregnancy outcome. A total of 7090 women completed the survey, and 34 women who collectively had experienced 68 pregnancies were subsequently interviewed.

**Results** Medication prescribing and use during pregnancy was common. The prescribing, dispensing and taking of some advised medications were restricted through women's or prescribers' fear of fetal harm. Lack of adherence to national prescribing guidance, conflicting professional opinion and poor communication resulted in maternal anxiety, avoidable morbidity and women negotiating complex and distressing pathways to obtain recommended medications. In contrast, some women felt overmedicated and that pharmacological treatments were used without exploring other options first.

**Conclusion** Increased translation of national guidance into practice and greater personalisation of antenatal care are needed to improve the safety, efficacy and personalisation of prescribing in pregnancy.

## BACKGROUND

Safe and effective prescribing is an essential component of antenatal care. Prescribing in pregnancy requires additional knowledge and caution due to the potential for teratogenesis, altered pharmacokinetics, maternal concerns[1] and potential for short and longer term harm to the fetus.[2 3] Antenatal information[4] provided to pregnant women states that most medications reach the fetus and

## STRENGTHS AND LIMITATIONS OF THIS STUDY

⇒ The study gained new insights into the seldom researched topic of medication use in pregnancy.
⇒ A sampling frame ensured inclusion of women with social and medical complexities from across the UK.
⇒ Women without sufficient spoken English or internet access were excluded.
⇒ Interviews were conducted only with women, not the health professionals who prescribe.

all medication use should be discussed with health professionals. Many women discontinue or avoid medications in pregnancy,[5] however, lack of treatment can potentially have severe consequences.[1] Despite selective avoidance, medication use in pregnancy is common, with around 87% of pregnant women in the UK reporting medication use for short-term or chronic conditions.[6] Excluding vitamin and iron supplements, frequently prescribed and over-the-counter (OTC) medications taken during pregnancy include antacids, analgesics, anti-emetics and antibiotics.[6]

There are few medications that should ideally not be used by pregnant women due to their teratogenicity, for example, thalidomide, sodium valproate and isotretinoin. Most other medications are safe and widely used during pregnancy, and there are several national prescribing guidelines[7 8] for treatment during pregnancy. For example, the National Institute for Health and Care Excellence (NICE) guidance on antenatal and postnatal mental health recommends that health professionals should discuss the potential benefits of psychological interventions and psychotropic medication, the possible consequences of no treatment and

possible harms associated with treatment, and what might happen if treatment is changed or stopped, particularly if psychotropic medication is stopped abruptly.[8] For mild to moderate mental health problems, psychological interventions should form first-line management,[8] but access is limited by long waiting lists and high referral thresholds, while a lack of continuity in antenatal care leads to difficulty developing trusting therapeutic relationships.[9] Suicide is the leading cause of direct maternal death in the first postnatal year,[10] and reports have highlighted that treatments for depression may often be discontinued in pregnancy.[5 11]

The Royal College of Obstetricians and Gynaecologists have set out clear guidance on the diagnosis and subsequent management of nausea and vomiting in pregnancy across community, ambulatory day-care and inpatient settings, including a medication escalation ladder for use when first-line recommended treatments are ineffective.[7]

There is growing concern that public health messages aimed at pregnant women, including those relating to medication use, do not always fully reflect or explain the evidence base underpinning them and the nuances and complexity of information is lost.[12] The inability to receive effective medications is not without consequence for women and babies. A survey in 2015 identified that women with severe hyperemesis gravidarum frequently had difficulty obtaining swift treatment and support for debilitating pregnancy sickness resulting in some terminating an otherwise wanted pregnancy.[13] Likewise, although mental health problems are common during pregnancy,[14] previous reports have highlighted barriers to access appropriate treatment.[15]

The 'WRISK project: understanding and improving the way risk in pregnancy is communicated to women' was established to explore women's experiences of maternity-related public health and risk messaging, including those relating to medication use in pregnancy. The study aimed to hear women's voices through public involvement, quantitative and qualitative methodologies with a specific objective of exploring the views of women previously identified as feeling stigmatised or poorly served by current practice.[16] This paper focuses specifically on women's experiences of medication use in pregnancy as reported in open free-text comments in the survey and in-depth interviews.

## METHODS
### Study design
The study design consisted of an online survey and nested in-depth interviews with a subsample of survey participants. The survey was open to UK residents aged 16–45 years, who were pregnant or who had been pregnant in the last 5 years, regardless of pregnancy outcome. The survey was publicised through social media platforms including those of the study's charity partners. We invited survey respondents to express interest in further research involvement and recruited participants for the interviews

from this population. As the survey was exploratory, an a priori sample size calculation was not required.

For subsequent in-depth interviews, we used a sampling frame to ensure that we heard the voices of women at greater risk of the most severe pregnancy outcomes, maternal death or stillbirth.[11 17] The sampling frame aimed to ensure that 30–35 women were interviewed and included a minimum of 20% who were eligible for means-tested state benefits, a minimum of 20% from Black, Asian and Minority Ethnic backgrounds, and at least 6 women were interviewed with experience of the following pregnancy experiences: pregnancy <20 years of age, body mass index >30 kg/m$^2$, antenatal mental health problems or experience of hyperemesis gravidarum. Interested respondents who met the sampling criteria were selected using a random number table, until required numbers were obtained.

### Data collection
#### Survey
A participant information sheet and consent form were integrated into the survey. The survey included questions on participants' experience of the advice, information and support they received from different sources during pregnancy, with the focus on the respondent's most recent pregnancy. Questions with open free-text responses were included alongside questions with Likert scales and multiple-choice responses. There were 22 questions in total, two of which asked for open free-text responses and are included in this analysis. The two questions asked were as follows:

> Please use this space to let us know about your experience and about any areas where you feel advice, information, and support for decision-making in pregnancy could be improved. and

> Is there another issue you would like to tell us more about? If so, please tell us what this is.

We describe the survey and demographic characteristics of respondents in full in a previous publication.[16] For the purposes of this publication, we present free text responses from the survey pertaining to medication use in pregnancy only.

#### One-to-one interviews
The participant information sheets, consent forms and topic guides for the interviews were developed in collaboration with the combined researcher and charity project advisory group. Narrative topic guides allowed participants to tell their pregnancy stories and were tailored to various pregnancy outcomes (see online supplemental file 1). Participants were asked to primarily discuss their most recent pregnancy and were asked about antenatal prescribing and use of over-the-counter medications, among other topics.

Participants were offered the option of in-person or telephone interviews. Interviewers (RB and HT) made detailed field notes following each interview. Participants

were offered, and all received a high street vouchers worth £20. Audio files were transcribed verbatim using a commercial transcription service. Electronic transcripts were stored separately to any identifiable data on a secure IT system, and audio files deleted once they had been checked for accuracy.

## Research team

The trained, all female research team had extensive experience of working in pregnancy-related research and practice including social sciences (RB and HT), medical law (JS), public health (RB, JS and HT) and clinical midwifery (JS). Interviews were conducted and analysed by RB and HT.

## Terminology

The WRISK project was inclusive of all people who had experienced pregnancy in the previous 5 years. The project team always referred to individuals according to their self-determined gender. In this paper, we use the words 'woman/women' as the vast majority of participants self-identified as women. We use the term 'BAME', but acknowledge that this is problematic and present data on participant sociodemographics on a more granular level.[18]

## Data analysis

The interviews and free text comments in the survey were analysed thematically following Braun and Clarke's method.[19] Transcripts were coded and analysed using Dedoose[20] by RB and HT. All transcripts and accompanying data such as researcher's field notes were read in detail several times by both interviewers, and high-level codes pertaining to the original research questions were identified. Further codes were identified in the data during an inductive process, resulting in a coding framework informed by the data itself. Narrative summaries of data pertaining to each code were produced and cross-checked by RB and HT, and subsequently organised into high-level themes. We used Excel and STATA SE 15 to generate descriptive statistics on the survey and interview populations.[21 22]

## Patient and public involvement

The project oversight group included representatives from five diverse maternity user groups, healthcare professionals and researchers in addition to the study team. The group met regularly throughout the project from conception to dissemination and collectively informed all aspects of study design and delivery.

## RESULTS

The survey was completed by 7090 women, of whom 3175 (44.1%) expressed willingness for further involvement in research, and 34 were subsequently interviewed. The two questions which asked for open free-text responses and included in this analysis were completed by 2197 and 737 participants, respectively. Sociodemographic characteristics of all survey respondents and quantitative survey findings are published elsewhere.[16]

Interviews were carried out by RB and HT between July and November 2019, lasting approximately 45–60 min. Two pilot interviews were conducted in-person in April 2019 and were deemed to be of sufficient quality to be included in the analysis. All subsequent interviews were conducted by telephone and recorded using a dictaphone. The 34 women interviewed included those with experience of: a BMI>30 (n=7); antenatal medication use for mental health conditions (n=6); medication for hyperemesis gravidarum (n=9); age <20 years during pregnancy (n=7); or having a termination due to a perceived or actual risk either to themselves or their baby (n=7). Some participants fitted more than one category, while five had none of these experiences.

Sociodemographic characteristics of those interviewed are described in table 1. Quotes that are followed by a unique identifier, for example, (WRI…) are from interview participants, and quotes that are from survey respondents are clearly identified.

Four themes were identified in relation to medication use in pregnancy: 'fear of medications and self-regulation', 'feeling overmedicated', 'conflicting opinions' and 'running the gauntlet'. Some themes related to women's experiences of medication use in general, while others were condition specific.

### Fear of medications and self-regulation

Fear or anxiety for the potential for fetal harm caused by OTC and prescribed medications was expressed by women. They also reported prescribers' reluctance to prescribe for this reason. Participants expressed support for the precautionary approach that medications should be kept to a minimum during pregnancy. The fear of causing fetal harm through taking medication resulted in women reducing medication even when this was required to control serious medical conditions.

Two women described the consequences of stopping prescribed medication for pre-existing medical conditions. One woman stopped taking her asthma medication, which resulted in hospitalisation:

> I stopped taking my inhaler, just because I just didn't want to harm the baby. I took a bad cold, and I was coughing up blood, and then they sent me to hospital for all these other tests. They thought I had a clot or something. So that obviously scared me. Then I just said, screw it. I'm just going to take my asthma medication, and that was that. [WRI28]

Similarly, another woman experienced a serious migraine and temporary blindness, following discontinuation of medication for cranial hypertension:

> I have a brain condition called… Well, it's cranial hypertension, so it just means that my body produces too much spinal fluid, which pushes on my

**Table 1** Sociodemographic characteristics of interview participants at time of interview

| Sociodemographics | N (34) | % |
|---|---|---|
| **Age (years)** | | |
| 19–20 | 2 | 5.9 |
| 21–25 | 9 | 26.5 |
| 26–30 | 7 | 20.6 |
| 31–35 | 7 | 20.6 |
| 36–40 | 7 | 20.6 |
| 41–45 | 0 | 0 |
| 45+ | 1 | 2.9 |
| Missing | 1 | 2.9 |
| **Highest level of education** | | |
| Secondary school | 2 | 5.9 |
| Apprenticeship/HND/NVQ | 3 | 8.8 |
| A-levels | 6 | 17.6 |
| Undergraduate degree | 13 | 38.2 |
| Postgraduate degree | 10 | 29.4 |
| **Relationship status** | | |
| Married | 23 | 67.6 |
| Have a partner and live with them | 9 | 26.5 |
| Have a partner and live separately | 1 | 2.9 |
| Polyamorous | 1 | 2.9 |
| **Ethnicity** | | |
| White: English/Welsh/Scottish/Northern Irish/British | 24 | 70.6 |
| Black/African/Caribbean/ Black British: African | 3 | 8.8 |
| Mixed/multiple ethnic groups: White and Black African | 1 | 2.9 |
| Asian/Asian British: Chinese | 1 | 2.9 |
| Asian/Asian British: Indian | 1 | 2.9 |
| Black/African/Caribbean/ Black British: Caribbean | 1 | 2.9 |
| Mixed/multiple ethnic groups: White and Black Caribbean | 2 | 5.9 |
| Mixed/multiple ethnic groups: White British and Middle Eastern | 1 | 2.9 |
| **Gender** | | |
| Female | 33 | 97.1 |
| Non-binary | 1 | 2.9 |
| **Receive state benefits?** | | |
| Yes | 10 | 29.4 |
| No | 23 | 67.6 |
| Missing | 1 | 2.9 |
| **Pregnancy history** | **Mean** | **Range** |
| Gravidity | 2 | 1–5 |
| Live births | 1.2 | 1–4 |
| Terminations/abortion | 0.3 | 0–2 |
| Miscarriage/stillbirth | 0.3 | 0–2 |

brain, which causes migraines, essentially. … because there isn't a lot of research into this condition in pregnancy, I decided, personally, to stop taking the medication. Yes, so I stopped all my medication and, thankfully, touch wood, everything was fine. … I did struggle with a few migraines throughout the pregnancy, one which was quite severe, which resulted in a loss of vision in my left eye, but luckily, it

came back. So that was fine. That was quite difficult to deal with, though. [WRI23]

Self-regulation of medication particularly related to the use of analgesics. Women described conflict between their understanding of some medications being safe to take in pregnancy, with their desire to try 'not to take them' [WRI11] due to the possibility of potential risks to the fetus.

… we all know it's (paracetamol) is generally safe, it's one of the safest painkillers there is, so it's the only one you can take, but even just taking that, women don't like- you know, they have a headache instead of taking the medicine, because they feel so guilty. [WRI30]

Another woman described managing with paracetamol for pelvic girdle pain even after she had been prescribed codeine-based medication:

Yes, one of those [codeine-based analgesia] and I was reading the side effects and I was like, 'No, I'm not happy taking it and having that go into my baby.' I just literally was taking paracetamol and I was just trying to take the lowest dose of paracetamol that I could really. [WRI4]

While women generally tried to avoid prescribed medication, recommended vaccines, including for influenza, appeared to be accepted. Supplements were viewed as positive and often taken without any apparent fear of causing an adverse pregnancy outcome.

Interviewer: Did you take medications while you were pregnant?

No, not at all.

Interviewer: No, not at all. Okay.

Beyond supplements and the influenza jab and recommended stuff like that. I had a whooping cough jab, I think, with [daughter's name] but no regular medication of any sort. (WRI7)

There were many examples of health professionals reinforcing the message that medication should be avoided in pregnancy.

Throughout my pregnancy I really struggled with my mental health. So, at one point, I don't know how pregnant I was, I must have been about six or seven months, we got an emergency appointment at my GP's surgery… and we said, 'I'm really struggling. Is there anything you can do?' His first reaction was that I shouldn't be on any medication because I was pregnant. [WRI29]

Health professionals expressing their own concerns around prescribing had a particularly negative impact on women. One interviewee [WRI17] explained her concern when a hospital consultant prescribing "put his hands together in prayer and said, '*God, forgive me for giving you*

*this. I hope your baby's okay'*" when prescribing ondansetron for severe hyperemesis gravidarum (HG).

Other women had similar experiences when accessing medication for their mental health. WRI33 said:

Because I was being so sick, I went to the doctor's about anti-sickness medication. She saw I was on antidepressants and she made me feel like I was the world's worst mother. I don't know whether she meant to, but she was just worried that, 'Ooh, you're pregnant and you're on medication.' She made me feel like, 'You're hurting your baby'. Then when I went to the pharmacy to pick up my meds, they made me feel like it on the same day as well. I think that was the point where I started to really decline in my mental health. It just seemed to trigger something off that I was going to be a terrible mother. [WRI33]

### Feeling overmedicated

Some participants reported feeling overmedicated and that pharmacological treatments were used without exploring other options first. This was the case for women with both mental and physical health concerns:

I feel like they push drugs on people too quickly, that is just the go-to. I had a traumatic birth with number two. That's why I went on the antidepressants with number three, because I was terrified of giving birth. They just kept telling me baby was getting so big, blah blah blah. I was, literally, terrified of giving birth to her. They just put me on antidepressants. There was no, 'Let's talk about it, get counselling, and get over the traumatic birth.' It was just mask it really. I don't think that's right to do when you're dealing with, obviously, a pregnant woman, a baby. [WRI31]

Current guidance relating to gestational diabetes recommends that metformin is introduced if good blood glucose control is not achieved with 1–2 weeks of dietary change and exercise.[23] One survey respondent would have preferred to extend this duration before commencing medication.

I would have liked to have been better involved in decision making around my options for treatment in relation to my gestational diabetes… I ended up with 2 tablets of metformin which I think was overkill - I would have liked to have had the opportunity to try harder with diet and exercise before starting medication. (survey respondent)

### Conflicting opinions

Women who required prescribed medication in pregnancy described frustration and distress resulting from conflicting opinions of health professionals. This conflict was particularly felt by women with hyperemesis gravidarum or mental health problems. One woman who required ondansetron for severe hyperemesis gravidarum was prescribed it, later to have this decision questioned by another general practitioner (GP) in the same practice:

So, I was prescribed by one doctor the ondansetron and because I was running out, they said I had to speak to another doctor just to check it was obviously working, just a check-up on it, to get the repeat prescription. Then when this doctor phoned me, she said, 'You shouldn't be taking this. This is for cancer patients. There's no reason why you should be taking it. You're putting your baby's life at risk'. … She just wasn't having any of it. I even told her that I'd just had a scan like 45 minutes before the phone call, and she said, 'Oh well, that can change at your 20 week scan. Don't come crying to me when they tell you that your baby has got heart defects'. [WRI15]

Another participant described the distress that resulted when a pharmacist would not fulfil a prescription for anti-depressants prescribed by her GP.

They just wouldn't give me my prescription, and people were behind me in the queue and could hear what I was on, that I was pregnant. I wasn't 12 weeks yet, so I hadn't told anybody. I had to just sit there waiting for them to decide whether I could have it or not and then they said, 'No, you can't have it.' I left without any medication, and I was just crying in the car. [WRI33]

Two interview participants said that conflicting and contradictory advice from healthcare professionals was 'staggering'.

I found through everything in my pregnancy that all the doctors I saw had completely different advice and completely different opinions to each other… [WRI6]

The conflict was seen across all professional groups. One respondent described conflicting information from her midwife and GP.

I was advised by my GP to continue my anti-depressant medication. My midwife continually questioned this decision throughout the whole pregnancy. (survey respondent)

### Running the gauntlet

This theme particularly related to prescribing for hyperemesis gravidarum. Some women experienced prolonged periods prior to getting effective treatment. Four women who were suffering with hyperemesis gravidarum described their experience of the current guidance.

they tried me on cyclizine, which just made me dizzy and did nothing for the sickness. I went very dizzy and sleepy. And then Stemetil… Which also had no effect. So, I was, kind of, getting desperate. And then, … they gave me ondansetron which is what they give to chemo patients, you probably know, and it was like I was alive again. … They were sympathetic, but I didn't get a lot of advice on what effect it would have on the baby. Just, you know, like, with the Cyclzine

and the Stemetil, they were very much, like, 'It's been used a lot and we can be certain from observations that it won't have any effect.' So that was fine. They were, kind of, reluctant to, like, move me onto the stronger stuff. [WRI1]

… they could see that I couldn't even take a sip of water without being sick, so I needed something. So, first off, I was given a tablet that you put between your gum and your teeth. It wasn't working. So, I went back and I got something called cyclizine, which is a tablet. That started to make it a little bit better for a couple of weeks and then I just got worse again. So, I had to go back again and …(they) gave me a pill called ondansetron. Then that was working, but I was still quite bad. So, I went back again and they gave me a combination of cyclizine and ondansetron, and that did the trick in the end. [WRI6]

Even when women were prescribed treatment for hyperemesis, they could then be discouraged from taking it by the prescriber:

The most unhelpful piece of advice I received was from a doctor, who prescribed me anti-sickness tablets for hyperemesis but said 'try not to take them unless you feel you need them' because 'we can't say that any medicine is safe in pregnancy'. I was very unwell, very dehydrated and very disoriented and was afraid of causing harm to the baby so I didn't take them. (survey respondent)

Another woman who was prescribed medication received comments from a pharmacist who suggested they may be unnecessary:

My husband went to collect my prescription when I was discharged, and was told that I was being 'irresponsible' and 'putting the baby at risk' by taking the medication. Later in pregnancy (I had HG until the moment of delivery), another pharmacist told me that I shouldn't still be taking medication as morning sickness would've stopped by my late stage, implied it was all in my head, and suggested that I try a herbal remedy instead. (survey respondent)

### DISCUSSION

This study adds important understanding to the seldom explored topic of women's experiences of antenatal medication use. Many women wished to reduce exposure to OTC and prescribed medication use during pregnancy, but dietary supplements and vaccines were generally accepted by pregnant women. This contrasted to treatments for mental health problems and hyperemesis gravidarum, conditions where pregnant women described having to negotiate conflicting opinions of health professionals to obtain recommended or effective treatments. Too often prescribing was more restrictive than recommended in national guidance resulting in

avoidable, or prolonged, maternal morbidity, distress and anxiety. Similar to the lack of preconceptual and antenatal care for women with epilepsy highlighted in reviews of maternal deaths,[11] where information on medications used for chronic conditions was not shared, some women discontinued treatments without medical consultation resulting in hospitalisation or exacerbation of symptoms.

In the UK, prescribing for pregnant women is undertaken by different health professionals, which complicates communication. Women with existing medical conditions, including epilepsy, require preconceptual advice on medication use.[24] GPs are frequently involved in prescribing decisions, and commonly prescribed medications such as iron therapy and low-dose aspirin may be prescribed and administered by midwives. Women with additional obstetric needs will receive care led by obstetricians, who may share prescribing with GPs or specialist physicians. This arena is further complicated by public health bodies who are responsible for producing public health risk messages for pregnancy being independent from both primary and secondary care where prescribing occurs. This multidisciplinary approach to antenatal prescribing was found often to be fragmented, with women hearing conflicting opinions even from different members of the same professional group. The role of pharmacists as medication 'gatekeepers' and their refusal to dispense prescribed medications highlights the importance of their inclusion in system improvements.

Even when effective treatments were endorsed in national guidance, challenges in implementation were found. Despite RCOG guidance on the management of hyperemesis gravidarum,[7] women reported being denied access to effective treatments. Ensuring health professionals have easy access to up-to-date guidance on specialist aspects of care and sufficient time and support to incorporate guidance into their practice is imperative.

While prescribers need to balance maternal benefit with potential fetal harm when prescribing in pregnancy, women's individual circumstances were not always considered, and they were not fully engaged in decision making. Possibly reflecting the tendency of health professionals to overestimate the teratogenic potential of drugs,[25] we found many examples where health professionals used fear of fetal harm to justify a refusal to prescribe or dispense otherwise recommended medications. This had a significant impact not only on women's health, but on their emotional well-being with one reporting they were made to feel like the 'world's worst mother'.

Some women felt antidepressants and metformin for gestational diabetes were offered in preference to non-pharmacological options. This may reflect that the availability of talking therapies does not meet demand, or in relation to gestational diabetes, a lack of informed personalised conversations on the effectiveness of methods to obtain glycaemic control.

High-quality, easily accessible information on the safety of medicines is available[26] but appeared to be underutilised in informing individualised discussions. Better

reporting systems on outcomes related to medication use in pregnancy are needed, including making the results of clinical trials more generalisable through, where appropriate, the inclusion of pregnant women.[27] Minimising prescribing in pregnancy became deeply entrenched in the ethos of antenatal care following the thalidomide tragedy.[28] The belief that all medication use during pregnancy carries risk is commonly held among women, with paracetamol, antibiotics and antidepressants[29] considered to be on a continuum of increasing risk. The potential for this position to cause harm is increasingly being realised. Some women with epilepsy and others with serious mental health conditions have died because of an over stringent position on medication avoidance.[11] More recently, the numbers of pregnant women who died during the COVID-19 pandemic could have been reduced through earlier acceptance of the safety of vaccination in pregnancy, better public health communication and greater use of effective treatments among those seriously ill with COVID-19.[30] While the need to reduce maternal mortalities through improved prescribing is already appreciated, our study suggests that physical and mental morbidity caused through the lack of access to effective treatments for mental health conditions and hyperemesis gravidarum is likely to be very common and requires improvement.

Strengths of this study include adding to knowledge on a seldom explored topic, the geographical spread and high number of survey participants from across the UK, and use of a sampling frame for the selection of interview participants. A weakness was that the sampling frame design may have shaped our findings and focused attention on certain experiences of medication use at the expense of others of equal concern to women. Our survey was self-selecting and may reflect the views of those more motivated to participate in research. The survey was only available in English and via the internet, excluding some groups from participation including non-English reading women and those with limited or no access to the internet. This study would have been strengthened by the inclusion of clinicians to better understand their attitudes and understandings of prescribing and dispensing during pregnancy.

## CONCLUSION

Medication prescribing and use during pregnancy is common. However, fear of fetal harm restricts the prescribing and taking of some advised medications. Analgesics were commonly avoided or taken at lower than the therapeutic or prescribed dose and some women reduced or discontinued medications for chronic conditions without medical oversight. Where existing clinical guidance was not followed, or there was conflict in professional opinion, reluctance to prescribe or dispense resulted in women needing to negotiate complex and distressing pathways to obtain required medications. The study identified aspects of antenatal prescribing where

improvement in knowledge, communication or practice is required to ensure maximisation of the safety, efficacy and personalisation of prescribing in pregnancy.

**Author affiliations**
¹School of Healthcare Sciences, Cardiff University, Cardiff, UK
²Centre for Reproductive Research & Communication, British Pregnancy Advisory Service, London, UK
³Pregnancy Sickness Support, Bodmin, UK
⁴Department of Primary Care and Population Health, University College London, London, UK
⁵School of Social Sciences, Cardiff University, Cardiff, UK

**Acknowledgements** We acknowledge and thank the group of over 7000 women who shared their intimate stories of pregnancy with us. Special thanks go to the Oversight Committee: Dame Cathy Warwick, BPAS Chair of Board of Trustees and Chair of the Oversight Committee; Rebecca Brione, formerly of Birthrights; Jane Fisher, Antenatal Results and Choices; Caitlin Dean, Pregnancy Sickness Support; Elizabeth Duff, NCT; Professor Irene Petersen, University College London; Amber Marshall, BigBirthas.co.uk; Professor Fiona Woollard, Southampton University; and the rolling representatives from Public Health Wales. We dedicate this paper to the late Dr Heather Trickey, without whom the project would never exist. Thank you for your inspiration and dedication to the health and wellbeing of all women.

**Contributors** CM and HT conceptualised the study and secured funding. CM, RB, HT, JS, IP and CD developed the protocol. HT and RB conducted the data collection and analysis. CM, JS, HT and RB provided study management. JS drafted the original manuscript. RB, IP and CD contributed to writing. CD and IP were members of the project advisory group. All authors approved the final manuscript. JS and RB are the guarantors of the work.

**Funding** This work was supported by the Wellcome Trust as part of the WRISK Project (212089/Z/18/Z).

**Competing interests** None declared.

**Patient and public involvement** Patients and/or the public were involved in the design, or conduct, or reporting, or dissemination plans of this research. Refer to the Methods section for further details.

**Patient consent for publication** Not applicable.

**Ethics approval** This study involves human participants and ethical approval was granted by the Research and Ethics committee of the School of Social Sciences at Cardiff University SREC/3201. Participants gave informed consent to participate in the study before taking part.

**Provenance and peer review** Not commissioned; externally peer reviewed.

**Data availability statement** Data are available upon reasonable request. Data are available on request.

**ORCID iD**
Rebecca Blaylock http://orcid.org/0000-0003-4317-1638

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
