## [Reviewer comments · BMJ Open]

ARTICLE DETAILS

TITLE (PROVISIONAL)	Women's experiences of over the counter and prescription medication during pregnancy in the UK: findings from survey free-text responses and narrative interviews.
AUTHORS	Sanders, Julia; Blaylock, Rebecca; Dean, Caitlin; Petersen, Irene; Trickey, Heather; Murphy, Clare

VERSION 1 – REVIEW

REVIEWER	McTaggart, Stuart Public Health Scotland
REVIEW RETURNED	02-Nov-2022

GENERAL COMMENTS	Thank you for the opportunity to review the paper "Try not to take them': Women's experiences of over the counter and prescription medication during pregnancy in the UK" by Sanders et al. The authors report on women's attitude to, and experience of medication use during pregnancy, including the advice given by and apparent attitudes of Healthcare Professionals. I considered the paper to be well structured and well written and it made good use of direct quotes from survey participants to illustrate lived patient experiences. The authors acknowledge that the self-selecting nature of the survey and sampling frame for interviews might have introduced bias but, nevertheless, I feel that it offers valuable, and at times concerning, insights into pregnant women's experience with accessing and taking medications and, indirectly, into their experience that HCPs do not always handle this topic well. I do have one comment that I feel should be addressed by the authors: Abstract, Results and Conclusion: The results section of the abstract opens with "Medication prescribing and use during pregnancy was common and some medications were prescribed and taken without concern." However, I could find nothing within the results that quantifies, or otherwise supported this statement. There are several general statements, but it is not always clear whether they are being made with reference to the full survey cohort or only among those interviewed. In my opinion, the results would benefit greatly from some quantification to support such statements as are presented in the authors already published paper describing the survey and the resulting themes of experience.
---

	The conclusion also opens with a similar statement and again this jarred with me as I did I did not feel that this paper had demonstrated such a conclusion to me. I also noted what would appear to be a couple of typos: Page 5, Line 14: '...consisting...' should read as consisted Page 11, Line 40: 'Health professions...' should read as Health professionals...
--	--

REVIEWER	Solomon, Daniel Brigham and Women's Hospital, Division of Rheumatology, Immunology and Allergy
REVIEW RETURNED	06-Nov-2022

GENERAL COMMENTS	The authors present an interesting mixed methods study regarding women's use of medications during pregnancy. This is a topic of great interest to the public where there is likely room for improvement in practice.  1. The mixed methods design is appropriate. However, I found that the authors focused on the interviews almost exclusively with relatively little analysis/results/discussion of the survey. While it is possible that the survey results were less instructive than the interviews, I anticipated more information being provided regarding survey results. 2. The quotes are very instructive and well chosen to illustrate the issues authors want to discuss. 3. The discussion is interesting and well written. I have the sense that this study and the discussion would nicely form the basis of a set of recommendations on improvements in prescribing during pregnancy. 4. However, a major concern raised is the conflicting (and sometimes inaccurate) information provided by clinicians about medication use during pregnancy. However, clinicians are not subjects in this research. It would have been instructive to interview or survey clinicians to better understand their attitudes (and misunderstandings) regarding prescribing during pregnancy. Clearly, recommendations for improvement will need to include clinicians, so this study would have been more impactful if they were included.
---

REVIEWER	Tran, Viet-Thi Paris Diderot University, Department of General Practice
REVIEW RETURNED	14-Nov-2022

GENERAL COMMENTS	Thank you for letting me review this qualitative study on women's experiences of over the counter and prescription during pregnancy. Overall, it was a pleasant read. The article is nicely written and interesting. I have only four minor comments:  1) I would clarify the fact that this study is a qualitative study. This would entail:  - expliciting the design (qualitative study) in the title - rewriting the abstract and focusing on the qualitative part (maybe one sentence stating that women were recruited from a larger survey) - rewrite the methods to focus on the qualitative part (for me the survey is not "study design" but related to the recruitment of
---

	participants). I would simplify (if not drop) the survey section of the methods). 2) I think the abstract does not support the text. besides focusing on the qualitative methods used (interviews, recorded, thematic analysis...), I would emphasize the main results from the text. As is, the abstract semmes much more related to the survey part...which is not reported in this article. 3) Table 1 is strangely formatted. Please adopt the presentation from "classic" research studies and avoid having different columns depending on the type of data. 4) In the results, it could be interesting to precise whether some answers seemd associated with participants' characteristics.
--	---

REVIEWER	Lauffenburger, Julie Brigham and Women's Hospital, Harvard Medical School, Division of Pharmacoepidemiology and Pharmacoeconomics
REVIEW RETURNED	15-Nov-2022

GENERAL COMMENTS	This is a clear multi-method study of surveys and interviews among women with recent pregnancy and medication use. There are some aspects that need to be clarified but otherwise appears to be a useful contribution to the literature. Several specific comments are outlined below. Major comments:  - The study seems to overall overemphasize the survey and also not clearly describe both modalities within the study. Please clarify particularly in the abstract that the data are both from surveys and qualitative interviews. - The conclusions are not fully supported by the data. In particular, the data from the qualitative interview do not suggest that personalization is actually necessary, just more that discussion needs to happen between patients and providers. Recommend re-framing the overall conclusions, especially in the abstract. Minor comments:  - More detail about how the women were identified from the surveys would be helpful. The response rate otherwise seems low. - How was the interview guide developed? More information on this would be helpful. - How many individuals were involved in the coding and was a 3rd party used to adjudicate any differences? More information on this would be helpful. - Table 1: Please collapse to only two columns and indicate the columns where they are mean (range) rather than N (%). - Suggest shorter quotes from fewer individuals to give a larger range of the types of responses. - Recommend a summary table of the key findings from the themes and surveys. Discretionary comments:  - The title idoes not seem to align with the main themes.
---

VERSION 1 – AUTHOR RESPONSE

Reviewer one	
--

I considered the paper to be well structured and well written and it made good use of direct quotes from survey participants to illustrate lived patient experiences. The authors acknowledge that the self-selecting nature of the survey and sampling frame for interviews might have introduced bias but, nevertheless, I feel that it offers valuable, and at times concerning, insights into pregnant women's experience with accessing and taking medications and, indirectly, into their experience that HCPs do not always handle this topic well.	Thank you.
Abstract, Results and Conclusion: The results section of the abstract opens with "Medication prescribing and use during pregnancy was common and some medications were prescribed and taken without concern." However, I could find nothing within the results that quantifies, or otherwise supported this statement. There are several general statements, but it is not always clear whether they are being made with reference to the full survey cohort or only among those interviewed. In my opinion, the results would benefit greatly from some quantification to support such statements as are presented in the authors already published paper describing the survey and the resulting themes of experience.	 - Thank you- we agree with this reflection and have removed this sentence from the abstract and conclusion. - The findings are derived from the 34 in-depth interviews and free-text responses which pertained to medication use. The suggestion to quantify our qualitative findings is more in-keeping with a positivist approach to qualitative research, which is contrary to our approach, and we are unable to repeat findings already published elsewhere.
The conclusion also opens with a similar statement and again this jarred with me as I did I did not feel that this paper had demonstrated such a conclusion to me.	We have removed the sentence with reference to some medications being prescribed and taken without concern.
I also noted what would appear to be a couple of typos: Page 5, Line 14: '...consisting...' should read as consisted Page 11, Line 40: 'Health professions...' should read as Health professionals...	Thank you, both typos have now been corrected.
Reviewer two	
The mixed methods design is appropriate. However, I found that the authors focused on the interviews almost exclusively with relatively little analysis/results/discussion of the survey.	Thank you for this comment. The findings from the survey have already been published elsewhere, hence we were unable to repeat them in this paper. We have ensured that we

While it is possible that the survey results were less instructive than the interviews, I anticipated more information being provided regarding survey results.	have clearly signposted this in the text (lines 218-219)
The quotes are very instructive and well-chosen to illustrate the issues authors want to discuss.	Thank you.
The discussion is interesting and well written. I have the sense that this study and the discussion would nicely form the basis of a set of recommendations on improvements in prescribing during pregnancy.	Thank you, we are involved in a number of policy initiatives which will make recommendations to improve women's experiences.
However, a major concern raised is the conflicting (and sometimes inaccurate) information provided by clinicians about medication use during pregnancy. However, clinicians are not subjects in this research. It would have been instructive to interview or survey clinicians to better understand their attitudes (and misunderstandings) regarding prescribing during pregnancy. Clearly, recommendations for improvement will need to include clinicians, so this study would have been more impactful if they were included.	We agree, and although we prioritised highlighting the experiences of women, we recognise that this is a limitation of our work. We have added in a sentence to this effect in the discussion (line 500-502).
Reviewer three	
Thank you for letting me review this qualitative study on women's experiences of over the counter and prescription during pregnancy. Overall, it was a pleasant read.	Thank you.
I would clarify the fact that this study is a qualitative study. This would entail: - expliciting the design (qualitative study) in the title - rewriting the abstract and focusing on the qualitative part (maybe one sentence stating that women were recruited from a larger survey) - rewrite the methods to focus on the qualitative part (for me the survey is not "study design" but related to the recruitment of participants). I would simplify (if not drop) the survey section of the methods).	Thank you. We have changed the title to:  - "Women's experiences of over the counter and prescription medication during pregnancy: findings from survey free-text responses and narrative interviews." - With reference to re-writing the abstract- we have clarified that the paper focuses on data from survey free-text responses and in-depth interviews. We cannot omit the survey entirely, as free-text responses were included in the analysis. - We have included more signposting in the methods section to make it clear that we are only drawing on the free-text responses from the survey plus interview data.

I think the abstract does not support the text. besides focusing on the qualitative methods used (interviews, recorded, thematic analysis...), I would emphasize the main results from the text. As is, the abstract seems much more related to the survey part...which is not reported in this article.	Thank you, with the amendments recommended above we think the methods now read more clearly. We have amended the results section of the abstract, and it is now a condensed description of the Results section. We have not named each theme, instead providing a coherent overall summary of our findings.
Table 1 is strangely formatted. Please adopt the presentation from "classic" research studies and avoid having different columns depending on the type of data.	Thank you, we have made the recommended changes to the table.
In the results, it could be interesting to precise whether some answers seemed associated with participants' characteristics.	Thank you for this suggestion. Given the small sample size of interview participants we would not be able to ascertain any meaningful, or indeed statistical, association between participants and their answers. Furthermore, the suggestion to quantify our qualitative findings is more in-keeping with a positivist approach to qualitative research, which is contrary to our approach.
Reviewer four	
This is a clear multi-method study of surveys and interviews among women with recent pregnancy and medication use. There are some aspects that need to be clarified but otherwise appears to be a useful contribution to the literature.	Thank you.
The study seems to overall overemphasize the survey and also not clearly describe both modalities within the study. Please clarify particularly in the abstract that the data are both from surveys and qualitative interviews.	Thank you, we have made amendments to the abstract and methods to more clearly state that the data are from free-text responses in the survey and qualitative interviews. Quantitative findings from the survey have already been published elsewhere and we have also signposted this in the text.
The conclusions are not fully supported by the data. In particular, the data from the qualitative interview do not suggest that personalization is actually necessary, just more that discussion needs to happen between patients and providers. Recommend re-framing the overall conclusions, especially in the abstract.	Our participants spoke clearly about their desire for prescribers to listen to and take account of their personal situations and preferences when prescribing. We would include such considerations with personalisation of care. The participants often described 'one way' conversations with prescribers presenting their opinion only. We agree that more discussion,

	i.e. two way conversations are required and desired by women.
More detail about how the women were identified from the surveys would be helpful. The response rate otherwise seems low.	We include the following information about how participants were identified from the survey in lines 140-148: “For subsequent in-depth interviews we used a sampling frame to ensure we heard the voices of women at greater risk of the most severe pregnancy outcomes, maternal death or stillbirth^{10 16}. The frame aimed to ensure the woman interviewed included a minimum of 20% who were eligible for means-tested state benefits, a minimum of 20% from Black, Asian, and Minority Ethnic backgrounds, and at least six women were interviewed with experience of the following pregnancy experiences: pregnancy <20 years of age, Body Mass Index >30Kg/m², antenatal mental health problems or experience of hyperemesis gravidarum. Interested respondents who met the sampling criteria were selected using a random number table, until required numbers were obtained.” 3,175 (44.1%) of survey respondents expressed willingness to be interviewed. We planned to interview 30-35 women in total and have added this in the description of our sampling frame in lines 142-143. Hopefully this makes it clearer that we did not have just 34 women who expressed an interest in being interviewed out of the 7,000+ survey participants, but instead capped the number of interviews within this range.
How was the interview guide developed? More information on this would be helpful.	The interview guide was developed in collaboration with our PPI group and other specialist organisations. Information to this effect is in lines 162- 167.
How many individuals were involved in the coding and was a 3rd party used to adjudicate any differences? More information on this would be helpful.	The coding was completed by RB and HT, and this has been added to the text in lines 191-192. A 3rd party was not used to adjudicate differences between their work. Braun and Clarke’s method of reflexive thematic analysis is

	a theoretically flexible approach to the analysis of qualitative data which enables the researcher(s) to identify themes within a given data set. It also recognises the researcher's own positionality and their role in producing knowledge as an asset rather than a form of bias. Within reflexive thematic analysis, unlike with positivist methods of analysis, there is no expectation that findings can be 'reproduced' or validated by another researcher.
Table 1: Please collapse to only two columns and indicate the columns where they are mean (range) rather than N (%).	Thank you, we have made the requested changes to the table.
Suggest shorter quotes from fewer individuals to give a larger range of the types of responses.	Apologies, but we are not clear how including quotes from fewer individuals would give a larger range of responses. Could the reviewer please clarify? We feel the quotes as currently presented do justice to the participants and their situations.
Recommend a summary table of the key findings from the themes and surveys.	This will be an editorial decision of BMJ Open and will of course include if required.

The title does not seem to align with the main themes.	We have changed the title to: "Women's experiences of over the counter and prescription medication during pregnancy: findings from survey free-text responses and narrative interviews."
--	---

VERSION 2 – REVIEW

REVIEWER	McTaggart, Stuart Public Health Scotland
REVIEW RETURNED	05-Dec-2022

GENERAL COMMENTS	Thank you for your revisions, which adequately address my previous comments. I noted one small typo in line 120 where the editing looks like it will result in The
---

	There is also some inconsistency in the separation of paragraphs and main text and comments i.e. line space / no line space, and sometimes multiple line spaces. That may simply be the result of the BMJ Open formatting process and will be resolved for final publication.
--	---

REVIEWER	Solomon, Daniel Brigham and Women's Hospital, Division of Rheumatology, Immunology and Allergy
REVIEW RETURNED	02-Dec-2022

GENERAL COMMENTS	Thanks for the revisions.
---------------------------

REVIEWER	Tran, Viet-Thi Paris Diderot University, Department of General Practice
REVIEW RETURNED	16-Dec-2022

GENERAL COMMENTS	I thank the authors for their revisions. Overall, I think only minor revisions are required 1) In the abstract, invert the two sentences of the participant paragraph (eligibility criteria should be stated before the number of participants) 2) Methods section, please precise the open-ended questions included in the survey 3) Results, please clarify the number of answers to the open-ended survey (how many responses and characteristics of respondents). If possible, in the results section clarify what data originated from interviews or open-text survey data. 4) Table 1 is still strangely presented, for example age has a 16-18 category with 0 participants. in the ethnicity group, there is "Black british" (?) "Asian British" (?) I would simplify this part.
---

REVIEWER	Lauffenburger, Julie Brigham and Women's Hospital, Harvard Medical School, Division of Pharmacoepidemiology and Pharmacoeconomics
REVIEW RETURNED	01-Dec-2022

GENERAL COMMENTS	The authors have satisfactorily addressed the comments.
---

VERSION 2 – AUTHOR RESPONSE

Reviewer 1	I noted one small typo in line 120 where the editing looks like it will result in Thhe There is also some inconsistency in the separation of paragraphs and main text and comments i.e. line space / no line space, and sometimes multiple line spaces. That may simply be the result of the BMJ Open formatting process and will be resolved for final publication.	Thank you, we have amended this typo.
------------	--	---------------------------------------

Reviewer 3	In the abstract, invert the two sentences of the participant paragraph (eligibility criteria should be stated before the number of participants)	Thank you, amended as requested.
	Methods section, please precise the open-ended questions included in the survey	Thank you. We have added this detail as requested and included the wording of the open-ended questions.
	Results, please clarify the number of answers to the open-ended survey (how many responses and characteristics of respondents). If possible, in the results section clarify what data originated from interviews or open-text survey data.	We have added the following text into the results section: “The two questions which asked for open free-text responses and included in this analysis were completed by 2,197 and 737 participants respectively. Sociodemographic characteristics of all survey respondents and quantitative survey findings are published elsewhere[16].” We have previously published the characteristics of the survey respondents elsewhere, and for brevity will not include an additional table in this manuscript.
	Table 1 is still strangely presented, for example age has a 16-18 category with 0 participants. in the ethnicity group, there is "Black british" (?) "Asian British" (?) I would simplify this part.	Thank you for highlighting this. We have deleted the row for the 16-18 category, and another row of missing data that was erroneously included. However, we will not be amending any of the ethnicity groups because these were derived from the national census categories and feel it is important to present them at this granular level.